# A Resistance Training Program on Patients with Liver Cirrhosis: A Randomized Clinical Trial

**DOI:** 10.3390/ijerph22081257

**Published:** 2025-08-11

**Authors:** Amanda Brown, Ingrid Dias, Jefferson da Silva Novaes, Leandro Sant’Ana, Bruno Perez Felix, Fabio Cahuê, Anderson Brito-Azevedo, Michel Silva Reis, Agnaldo José Lopes, Verônica Salerno, Roberto Simão, Estêvão Rios Monteiro, Renata de Mello Perez

**Affiliations:** 1Federal Institute of Southeast Minas Gerais—Campus Barbacena, Barbacena 36205-018, Brazil; 2Postgraduate Program in Physical Education (PPGEF), Universidade Federal do Rio de Janeiro, Rio de Janeiro 21941-599, Brazil; ingriddias.ufrj@gmail.com (I.D.); jeffsnovaes@gmail.com (J.d.S.N.); fabiocahue@gmail.com (F.C.); msreis@hucff.ufrj.br (M.S.R.); vpsalerno@yahoo.com.br (V.S.); robertosimao@ufrj.br (R.S.); e.rios.monteiro@souunisuam.com.br (E.R.M.); 3Strength Training Laboratory (LABFOR), Universidade Federal de Juiz de Fora, Juiz de Fora 36036-900, Brazil; leandrosantana.edufisica@hotmail.com; 4Undergraduate Program in Physical Therapy, IBMR University Center, Rio de Janeiro 22631-002, Brazil; bruno.felix@ulife.com.br; 5Postgraduate Program in Rehabilitation Science (PPGCR), Centro Universitário Augusto Motta, Rio de Janeiro 21032-060, Brazil; alopes@souunisuam.com.br; 6Liver Transplant Unit, Adventista Silvestre Hospital, Rio de Janeiro 22241-220, Brazil; andersonbrito.a@gmail.com; 7Postgraduate Program in Rehabilitation Science (PPGCR), Universidade Federal do Rio de Janeiro, Rio de Janeiro 21949-900, Brazil; 8Pulmonary and Tisiology Department, Pedro Ernesto University Hospital (HUPE), Piquet Carneiro University Policlinic (PPC), Universidade do Estado do Rio de Janeiro, Rio de Janeiro 20950-003, Brazil; 9Faculty of Medicine, Universidade Federal do Rio de Janeiro, Rio de Janeiro 21044-020, Brazil; renatamperez@gmail.com

**Keywords:** liver cirrhosis, physical exercise, muscle strength, quality of life

## Abstract

Background: Cirrhosis represents an advanced stage of hepatic fibrosis, and the metabolic changes resulting from liver dysfunction can cause impairment in physical capacity and quality of life. This study aimed to evaluate the effect of 12 weeks of resistance exercise on inflammatory markers, oxidative stress, physical conditioning, and quality of life in patients with liver cirrhosis. Methods: A total of 38 patients paired in the exercise (EG) and control (CG) groups participated in this study. The EG submitted to two weekly sessions of a program composed of resistance exercises. We evaluated the inflammatory markers (IL-6, IL-10, and TNF-α), oxidative stress (carbonylated protein, total thiols, enzyme superoxide dismutase, and catalase activity), body composition, handgrip strength (HGS), total volume of training, cardiorespiratory capacity (VO_2_, VCO_2_, and VE_peak_), exercise capacity, and quality of life. Results: Regarding the EG, when comparing the follow-up period to the baseline, significant differences (*p* < 0.05) were found in BMI, HGS, total training volume, cardiorespiratory capacity variables, the 6 min walk test, and quality of life. Improvements were observed, without significant differences, in the inflammatory profile, oxidative stress, and one of the markers of liver function. The CG showed a significant decrease (*p* < 0.05) in HGS and cardiorespiratory capacity after the intervention. Conclusions: In patients with liver cirrhosis, a resistance exercise program improved physical fitness and quality of life, without adverse events. In addition, it seems that this type of training can bring some benefit to the inflammatory profile and oxidative stress of this population.

## 1. Introduction

Cirrhosis is a disease of significant prevalence, and its complications contribute to high morbidity and mortality in affected individuals. Liver transplantation remains the sole therapeutic option for patients with end-stage liver disease. However, as highlighted by Debette-Gratien et al. [1], patients on the transplant waiting list often experience severe physical deconditioning, potentially leading to worse post-transplant outcomes. While physical exercise is recommended as a non-pharmacological approach for managing chronic diseases [2], the efficacy and safety of exercise in liver cirrhosis remain poorly studied and supported by limited evidence. Recent meta-analyses demonstrate that resistance exercise in combination with aerobic exercise reduces the incidence of serious events in patients with liver cirrhosis [3].

Patients with liver cirrhosis often exhibit dysregulation in inflammatory markers [4] and oxidative stress [5], which can further exacerbate the progression of the disease. Studies have shown elevated levels of pro-inflammatory cytokines [6] and reactive oxygen species [7] in patients with liver cirrhosis, contributing to the development of complications and impaired liver function. Understanding these underlying mechanisms is crucial for devising targeted therapeutic interventions to mitigate the impact of inflammation and oxidative stress in cirrhosis. The Systemic Inflammation Hypothesis [8] posits that Acute-on-Chronic Liver Failure arises from the exacerbation of pre-existing systemic inflammation and associated systemic circulatory dysfunction in acute decompensation. This exacerbation leads to organ failure(s) because of organ hypoperfusion and the direct deleterious effects of inflammatory mediators on organ microcirculation and cellular physiological homeostasis.

Resistance training has shown promising effects in studies by Bernardi, modulating inflammatory [9,10,11] and oxidative stress markers [9,11] among different special-conditions patients [12,13,14]. Several studies have observed decreases in pro-inflammatory cytokines and markers of oxidative stress after implementing a well-structured resistance training program [9,10,11]. These encouraging results indicate that regular resistance training could positively influence the inflammatory and oxidative status of patients with liver cirrhosis, potentially leading to improved health outcomes in this specific population [12]. While the exact mechanisms responsible for these responses are not fully elucidated, they might involve heightened antioxidant defenses [15], improve mitochondrial function [16,17], and reduce systemic inflammation [18,19]. Incorporating resistance training into the comprehensive care plan for cirrhotic patients may offer substantial therapeutic potential in managing the complications associated with chronic liver disease.

While there is a growing recognition of the importance of resistance training as an integral part of the treatment for patients with liver disease [12], the amount of available evidence is limited. Additionally, the improvement in inflammatory components in hepatic cirrhotic patients after resistance training still lacks a clear physiological explanation, highlighting the need for further investigations in this area. The plausible justification for conducting this study is the potential clinical benefit that resistance training can bring to patients with hepatic diseases. Due to the scarcity of robust evidence, it is essential to fill this scientific gap so that healthcare professionals can provide a more comprehensive and effective treatment approach for patients with hepatic diseases. Thus, this study aimed to evaluate the effect of 12 weeks of resistance exercise on inflammatory markers, oxidative stress, physical conditioning, and quality of life in patients with liver cirrhosis.

## 2. Materials and Methods

### 2.1. Participants

This study was conducted by a multidisciplinary team, and patient recruitment took place after several clinical visits at the hepatology center of the university hospital. A priori power analysis was conducted using G*Power software (version 3.1.9.6) with a medium effect size (f = 0.50), an alpha level of 0.05, and a statistical power of 90% (1−β = 0.90), indicating that a minimum of 47 participants would be required to detect significant between-group differences. Although the ideal sample size was not fully reached, a total of 38 patients diagnosed with liver cirrhosis were successfully enrolled, representing approximately 81% of the estimated statistical power. These individuals were randomly allocated into two groups, the Experimental Group (EG, n = 19) and the Control Group (CG, n = 19), as detailed in Figure 1 and Table 1. The allocation of participants ensured that the groups were matched in terms of liver disease etiology, severity, gender, and age. Exclusion criteria included (i) smoking, (ii) coronary artery disease, (iii) presence of hepatocellular carcinoma, (iv) use of vasoactive drugs (except propranolol), (v) hepatitis C treatment within the six months prior to intervention, (vi) corticoid use, (vii) ongoing infectious conditions, (viii) upper gastrointestinal bleeding within two weeks prior to the intervention, (ix) osteomyoarticular lesions, and (x) medium- or large-caliber esophageal varices without eradication found in upper digestive endoscopy.

Prior to this study, all participants received both verbal and written explanations of all study procedures and provided written informed consent to participate. Additionally, the participants completed the Physical Activity Readiness Questionnaire, and only those who answered NO to all questions were included in the study. All research procedures were conducted in accordance with Resolution No. 466/12 of the Brazilian National Health Council. The study protocol was submitted to and approved by the Institutional Review Board of the Clementino Fraga Filho University Hospital—Federal University of Rio de Janeiro (IRB number: 933923) and the Brazilian Registry of Clinical Trials—REBEC (RBR-54ngrx), and it adhered to the principles of the Declaration of Helsinki.

### 2.2. Experimental Procedure

This longitudinal study evaluated the patients at two time points, at baseline and after 12 weeks, and the experimental design was used to evaluate the effect of 12 weeks of resistance exercises on inflammatory markers, oxidative stress, physical conditioning, and quality of life in patients with liver cirrhosis (Figure 2). Participants visited the laboratory for 26 sessions during a 13-week period. The first two visits were used to familiarize all participants with all procedures. The remaining 12 weeks (24 visits) were used to assess, apply the resistance training, and reevaluate the participants. Patients with liver cirrhosis were allocated in EG (n = 19) and CG (n = 19) according to the etiology and severity of liver disease, gender, age, and resistance training program using the paired set condition consisting of alternating sets between two exercises for different muscle groups.

The development and progression of the program was adapted from the American College of Sports Medicine [20] recommendations for physical activity in older adults. EG exercises were performed in the following order: machine lat pull-down and leg press, bench press and leg extension, low row and leg curl, biceps curl and plantar flexion, triceps pulley, and abdominal. Before starting the session, participants performed a specific warm-up in the first exercises (machine lat pull-down and leg press) with 50% of the estimated overload to complete 10 to 12 repetitions. At the end of each session, stretching exercises were performed to the point of mild discomfort, for 5 min, for all major muscle groups (Figure 1). The total duration of each training session was approximately 40 min. All participants carried out the supervised program in groups of a maximum of 4 people. For each exercise, three sets of 10 to 12 maximum repetitions were performed, and in abdominal and plantar flexion exercises, volunteers should perform 15 to 20 repetitions with their body weight. An interval of 1 min was followed between the sets, and the load increments were performed whenever the patients exceeded the number of repetitions, considering the overload of the machine itself.

GC participants were instructed to maintain their daily activity routine during the 12 weeks of the protocol. All procedures were performed at the same time of day (in the morning) to avoid any confounding effect of circadian rhythm.

### 2.3. Measures and Procedures

*Body composition*: Body composition was analyzed using tetrapolar bioimpedance (INBODY^®^, model 230, Seoul, Republic of Korea) to measure weight, body mass index (BMI), lean mass, and fat mass. For greater accuracy, all volunteers were instructed to follow the preparing recommendation for the exam. This method of body composition assessment was deliberately selected due to its superior external validity when compared to more sophisticated techniques, aligning more closely with the tools commonly employed in clinical settings. Consequently, this choice enhances the ecological applicability and translational potential of the findings for routine clinical evaluation.

*Inflammatory Profile and Oxidative Stress*: The quantification of plasma cytokines interleukins (IL-6 and IL-10) and tumor necrosis factor alpha (TNF-α) was performed using the PicoKine ELISA kit (Booster, CA, USA) following the manufacturer’s instructions. The absorbance was measured at 412 nm in an ELISA microplate reader and the values are expressed as nmol of reduced DTNB/mg protein. The amount of Protein Carbonyls (PCs) in plasma is expressed as µMol/mL. Catalase activity in plasma was measured using the method described by Aebi [21] and is expressed in U/mL. Enzyme superoxide dismutase (SOD) activity was quantified using the protocol suggested by Misra and Fridovich [22], and the concentration is shown in U/mL.

*Muscle Strength*: The handgrip strength (HGS) assessment was performed by a closed hydraulic system that measures the amount of force produced by an isometric contraction. The test was conducted in the sitting position with the arm close to the body, elbow at 90°, and forearm in the neutral position [23]. The value used was the average of three measurements on the dominant side, with a 60 s rest period between sizes in position II of the Jammar^®^ dynamometer (Manhattan Beach, CA, USA).

*Neuromuscular Response*: The volume load (VL: ∑ (total number of repetitions x load for each exercise)) across the EG group condition was calculated [20].

*Cardiopulmonary Capacity*: The evaluation was carried out following the recommendations of Neder and Nery [24], on an electromagnetic braking cycle ergometer (Ergo-Fit^®^ 167 Cycle, Pirmasens, Germany) with the ramp type protocol. The volunteer was instructed to rest for 1 min while seated on the cycle ergometer. After this period, a 3 min warm-up was performed at a power of 15 W and a cadence of 60 rotations per minute (rpm). Finally, the physical exercise protocol started with increments of 10 W per minute and speed maintained at 60 rpm until the submaximal frequency (70% of HRmáx) or until physical exhaustion, that is, the inability to maintain the cadence or any symptom that limited the continuity of the test (dizziness, nausea, complex arrhythmias, and/or peripheral desaturation below 90%). The recovery period after the test consisted of 3 min at 25 W of power, followed by 3 min of passive rest after interrupting the pedaling. Ventilatory and metabolic variables were captured and recorded throughout the test period. The electrocardiogram (Wincardio USB) was monitored continuously throughout the experimental procedure. In the cardiorespiratory test, ventilatory and metabolic variables were obtained through a computerized gas analysis system (Medgraphics^®^, VO2000—Portable Medical System Corporation, São Paulo, Brazil) through which variables VO_2_, VCO_2_, and VE were calculated during maximum effort (PEAK).

*Exercise Capacity*: The 6 min walk test assessed exercise capacity. Patients were instructed to walk the longest distance along a corridor. An appraiser measured the time, and the total distance walked by the patient was recorded in meters [25].

*Quality of Life*: The quality-of-life assessment was performed using the Short-Form Health Survey questionnaire (SF-36). It is a multidimensional [26] instrument that assesses the patient’s perception of health status and its influence on quality of life. It consists of 36 questions divided into eight components. The evaluation was performed by assigning scores for each item which were transformed into a scale from zero to 100, where zero and 100 correspond to the extremes of worst and best quality of life, respectively [27].

### 2.4. Statistical Analysis

Fisher’s test for categorical variables was used to compare baseline characteristics of patients in both groups. The Shapiro–Wilk test was used to test parametric assumptions of normality. To test the effect of exercise on dependent variables, the paired *t*-test and the Wilcoxon test were used for data with normality not rejected or rejected, respectively. For ordinal variables, the Wilcoxon test was also applied. When necessary, the effect size of statistical comparisons was calculated by Cohen’s d formula. Effect size was considered small when d = 0.2, medium when d = 0.5, and large when d = 0.08. All analyses were performed using SPSS (v. 20) and GraphPad Prism (v. 6), with a significance level of 5% (*p* ≤ 0.05).

## 3. Results

One hundred and fourteen patients were recruited for this study, of which 37 did not meet the inclusion criteria, 31 refused to participate in the research, and 8 discontinued the intervention. The main reasons for refusing to participate were distance from home to the training location and the lack of independence for this displacement. Baseline characteristics were similar between EG and CG (Table 1).

### 3.1. Inflammatory Profile

Interleukins IL-6, IL-10, and TNF-α (Figure 3) were analyzed to investigate whether resistance exercise could improve the inflammatory profile of patients. At baseline and follow-up, there was no significant difference between the groups analyzed. However, it is noticeable that the EG exhibited a decrease in TNF-α values at the follow-up period compared to baseline (follow-up: 55.7 ± 47.2 pg/mL, baseline: 70.8 ± 58.8 pg/mL, *p* = 0.46, ES: 0.22), and an increase in IL-10 values at follow-up compared to baseline (follow-up: 6.4 ± 5.3 pg/mL, baseline: 5.2 ± 3.1 pg/mL, *p* = 0.25, ES: 0.25), indicating a slight improvement in the inflammatory profile. On the other hand, the CG showed no important modifications in TNF-α values at the follow-up time compared to baseline (follow-up: 51.9 ± 41.7 pg/mL, baseline: 43.5 ± 38.9 pg/mL, *p* = 0.34, ES: 0.16), and in IL-6 (follow-up: 6.4 ± 5.7 pg/mL, baseline: 5.7 ± 5.4 pg/mL, *p* = 0.11, ES: 0.08), suggesting a maintenance of the inflammatory profile. 

### 3.2. Oxidative Stress

To assess the impact of resistance exercises on oxidative stress, carbonylated protein levels, total thiol concentration, and the activity of SOD and catalase were examined (Figure 4). No significant differences were found between the analyzed groups at baseline and follow-up. However, in the EG, there was a trend towards decreased values of total thiols (1.7 ± 1.6 vs. 2.9 ± 1.7 nMol/mg protein, *p* = 0.02, ES: 0.57), carbonylated protein (6.7 ± 2.8 vs. 7.2 ± 5.2 μM/mL, *p* = 0.07, ES: 0.08), SOD (446.4 ± 20.0 vs. 450.1 ± 21.1 U/mL, *p* = 0.16, ES: 0.13), and catalase (2030.0 ± 2148.0 vs. 2849.6 ± 2698.4 U/mL, *p* = 0.60, ES: 0.25), suggesting a potential improvement in oxidative stress markers with resistance exercises. Conversely, in the CG, there was an increase in carbonylated protein levels (10.3 ± 6.8 vs. 8.4 ± 5.8 μM/mL, *p* = 0.10, ES: 0.23) and catalase activity (2283.2 ± 2322.6 vs. 1935.1 ± 1828.3 U/mL, *p* = 0.43, ES: 0.13) at the follow-up period compared to baseline.

### 3.3. Blood Analysis and Body Composition

Regarding blood analysis, no significant differences were observed between the follow-up and baseline periods in both the EG and CG. Aspartate aminotransferase, a marker of liver function, showed a slight improvement at the follow-up period compared to baseline in the EG, although without statistical significance (Table 2). In the body composition variables, a significant improvement in body mass index was observed in the EG at the follow-up compared to baseline. However, there were no significant differences in lean mass and fat mass between the two groups at the two time points (Table 2).

### 3.4. Muscle Strength

In HGS, a significant increase was found in the EG at the follow-up period compared with the baseline. The same variable in the CG showed a significant reduction between the follow-up and baseline periods. Concerning the total training volume, significant increases were found for all exercises in the follow-up period when compared to the baseline (Table 3).

### 3.5. Cardiorespiratory Capacity

Significant increases in VO_2_ peak were observed in the EG (1.30 ± 0.44 vs. 0.87 ± 0.30 L/min), VCO_2_ (1.46 ± 0.67 vs. 1.01 ± 0.55 L/min), and VE (36.56 ± 16.67 vs. 22.68 ± 7.63 L/min) at follow-up when compared to baseline. In the CG, the variables VO_2_ (0.80 ± 0.36 vs. 1.09 ± 0.47 L/min), VCO_2_ (0.77 ± 0.37 vs. 1.12 ± 0.48 L/min), and VE_peak_ (18.12 ± 8.19 vs. 25.30 ± 11.51 L/min) were significantly lower at follow-up compared to baseline (Figure 5).

### 3.6. Exercise Capacity

In the 6 min walk test, the EG showed a significant improvement in follow-up compared to the baseline, while in the CG, no differences were found (Table 3).

### 3.7. Quality of Life

Regarding quality of life, the EG group significantly improved the perception of physical functioning, the limitation of roles due to physical health, social functioning, role limitations due to emotional problems, bodily pain, and general perceptions of health at the time of monitoring when compared to the baseline. In the CG, no differences were observed in relation to the quality-of-life parameters (Table 4).

## 4. Discussion

This study represents a novel exploration of the effects of a resistance training program specifically tailored for patients with liver cirrhosis. The key findings of our research demonstrate significant improvements in both the inflammatory profile and oxidative stress markers, without any reported adverse effects.

To our knowledge, this is the first study investigating the impact of resistance training on interleukins (IL-6 and IL-10) and tumor necrosis factor-alpha (TNF-α) in this patient population. Although the differences did not reach statistical significance, there was a clear trend toward reduced TNF-α levels and increased IL-10 levels. Chronic liver diseases are often characterized by elevated levels of TNF-α, IL-6, and IL-10, with TNF-α being a key cytokine implicated in systemic inflammation [28,29]. Thus, the observed reduction in TNF-α levels, combined with an IL-10 increase in the exercise group, may suggest a shift toward a less inflammatory systemic state.

In patients with liver disease, a heightened inflammatory profile is frequently accompanied by increased oxidative stress [30]. This is manifested as elevated redox damage, diminished plasma antioxidant capacity, and reduced antioxidant enzyme activity in erythrocytes and liver tissue, all of which are hallmarks of liver cirrhosis [31,32]. The data from our study confirm the presence of significant oxidative stress in these patients. However, following the intervention, the exercise group exhibited lower levels of protein carbonylation and reduced plasma activity of the SOD and catalase, despite a decrease in total plasma thiol concentration. The activity of SOD and catalase is often interpreted as a compensatory mechanism in response to oxidative stress [33] and is associated with poorer outcomes in septic patients [34]. The observed reduction in oxidative stress, along with the improved inflammatory profile post intervention, supports the hypothesis that mitigating oxidative stress contributes to the clinical improvements seen in patients with liver cirrhosis, contributing to improving anabolic signaling and mitochondrial function. The intervention period (12 weeks) was lower than those necessary to significantly improve these biomarkers [35], but the trends found in our work can provide insights into other studies, with a greater time of intervention and sample size.

In terms of body composition, significant differences were observed only in body mass index, with no changes in lean or fat mass in either group. Previous research by Jones et al. [36] suggests that gains in lean mass typically do not become apparent until after 12 weeks of resistance training. Before such gains are observed, neural adaptations and enhancements in muscle fiber strength often predominate. Similarly, Aamann et al. [37] reported increased quadriceps cross-sectional area after a 12-week resistance training program, potentially attributed to differences in training volume. It is important to note that while both interventions lasted 12 weeks, the study by Aamann et al. [37] included 36 exercise sessions compared to our program’s 24 sessions. This variation in training volume could explain why lean mass changes were not detected in our study. It is conceivable that longer programs (>12 weeks) or those with a greater frequency of weekly sessions might yield more pronounced changes in lean mass.

A reduction in muscle mass and strength is a common finding in patients with advanced liver disease. Jones et al. [38] reported that these impairments seem to be independent of the cirrhosis etiology but are closely related to disease progression. In our investigation, we assessed muscle strength using handgrip dynamometry and observed a significant improvement in the group that underwent the resistance exercise protocol. Conversely, in the CG, who maintained their regular daily activities, strength declined during the follow-up period. Interestingly, the study by Hiraoka et al. [39] also utilized the same method to measure muscle strength, and their results align with ours. However, it is important to note that their training approach involved home exercises, making direct comparisons difficult.

According to the American College of Sports Medicine [20], muscle strength performance can be assessed by total training volume, which is calculated by multiplying the number of sets, repetitions, and load for each exercise. In the context of our review, no previous studies were found that employed the same methodology in the targeted population. Our results revealed a significant increase in training volume following the 12-week intervention with resistance exercises, potentially associated with enhanced dynamic strength. As mentioned earlier, in untrained individuals, strength gains may precede increases in cross-sectional area [38], suggesting that strength exercise alone could potentially mitigate the adverse effects of the disease on the musculoskeletal system. While not measured in this study, it is plausible to suggest that the improvements in muscle strength could be attributed to neuromuscular adaptations [40], including increased motor unit activation and improved coordination among recruited muscles. Moreover, it is conceivable that there might be an upregulation in protein synthesis and muscle hypertrophy, leading to greater muscle mass and, subsequently, enhanced strength gains [41]. Another hypothesis to consider is that resistance training may improve the efficiency of metabolic pathways involved in energy production, thereby enhancing anaerobic capacity and muscular endurance in these patients [42].

In our study, we noted a significant increase in cardiorespiratory capacity in the group that underwent the exercise program, whereas the CG showed a decline. While improvements in cardiorespiratory capacity are typically associated with aerobic exercises, in our investigation, we attributed the observed changes to substantial perfusion alterations in the patients. Moreover, resistance exercises have been shown to enhance oxygen extraction efficiency in skeletal muscle, which may have played a role in augmenting the cardiorespiratory capacity of the participants under study [43]. Additionally, the increase in muscle mass and strength may contribute to a reduced demand on the cardiovascular system during physical activity [44]. Moreover, the training might stimulate favorable adaptations in the autonomic nervous system, leading to a more balanced cardiac regulation and enhanced exercise tolerance [45].

The physiological complexities associated with this condition are associated with reduced levels of physical activity, which subsequently impact the daily activities of the affected population. To assess exercise capacity, we used the 6 min walk test, a tool that, as noted by Carey et al. [45], can serve as a predictor of mortality in patients awaiting liver transplantation. Our study participants demonstrated improved performance on the walk test following the intervention. These results lead us to propose that regular engagement in this type of exercise leads to gains in muscle strength and overall physical conditioning in these patients, ultimately influencing functional activities, such as walking.

The physiological complications of the disease are related to a decrease in the level of physical activity, with a consequent impact on the activities of daily living of this population. To analyze the exercise capacity, the 6 min walk test was developed, which, according to Carey et al. [45], can be used as a predictor of mortality in patients queuing for liver transplantation. Our volunteers showed an improvement in the performance of the walk test after the intervention. These findings lead us to believe that the regular practice of this type of exercise promotes gains in muscle strength and general physical conditioning in these patients, with implications for functional activities, such as walking. Given that exercise capacity is a recognized predictor of post-transplant outcomes, the observed improvements in functional performance following the intervention may have important clinical implications. Enhanced exercise capacity could contribute to better pre-transplant conditioning, potentially improving eligibility for liver transplantation and long-term prognosis.

Marchesin et al. [46] report that the clinical assessment of patients’ perception of health-related quality of life is a recent development. The same authors say that the traditional behavior based on important parameters to clinicians need to be integrated with patients’ opinions about their health status, reflecting how they feel and how much the disease affects their way of life. Although there is evidence to suggest that aerobic exercise improves quality of life, there is little evidence about the effect of resistance exercise. In the literature, Aamann et al. [37] found an improvement only in the mental health and vitality domains of the SF-36 questionnaire after the intervention with resistance exercises. Despite the similarities in the protocols, we observed a broader benefit of this type of exercise on quality of life.

Our study presents relevant results, but it is necessary to consider some limitations. The sample size was not large, but we included a sample size similar to that of other studies [1,37,47,48] that evaluated physical training in cirrhotic patients. Concerning body composition, we used a method considered less accurate, but the patient himself was his control, which contributed to compensating for the limitations of the method. Analysis of the oxidative stress and inflammatory biomarkers was made in plasma, which presents a high variability in cirrhotic patients. To mitigate errors in interpretation, the effect size of analysis was calculated to improve the analysis of data. In addition to these factors, our sample was composed of patients with compensated liver cirrhosis, which makes it impossible to extrapolate our results to decompensated patients. However, we understand the need for further studies, especially with patients with liver cirrhosis in more critical states of the disease.

## 5. Conclusions

A program composed exclusively of resistance exercises increases cardiorespiratory capacity, muscle strength, exercise capacity, and quality of life in patients with liver cirrhosis without adverse events. Also, this type of training seems to bring some benefit to the inflammatory profile and oxidative stress. Therefore, it should be considered a recommendation in the clinical follow-up of these patients.

## Figures and Tables

**Figure 1 ijerph-22-01257-f001:**
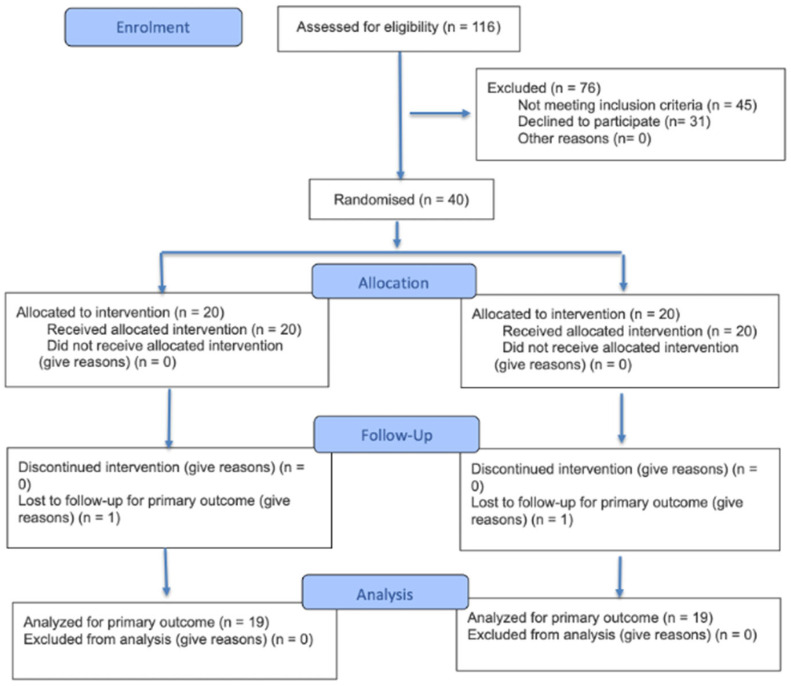
CONSORT flowchart.

**Figure 2 ijerph-22-01257-f002:**
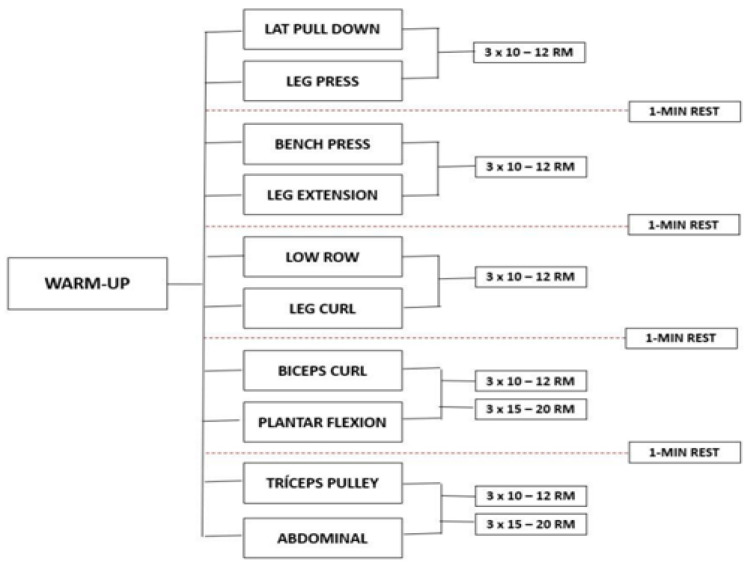
Study design.

**Figure 3 ijerph-22-01257-f003:**
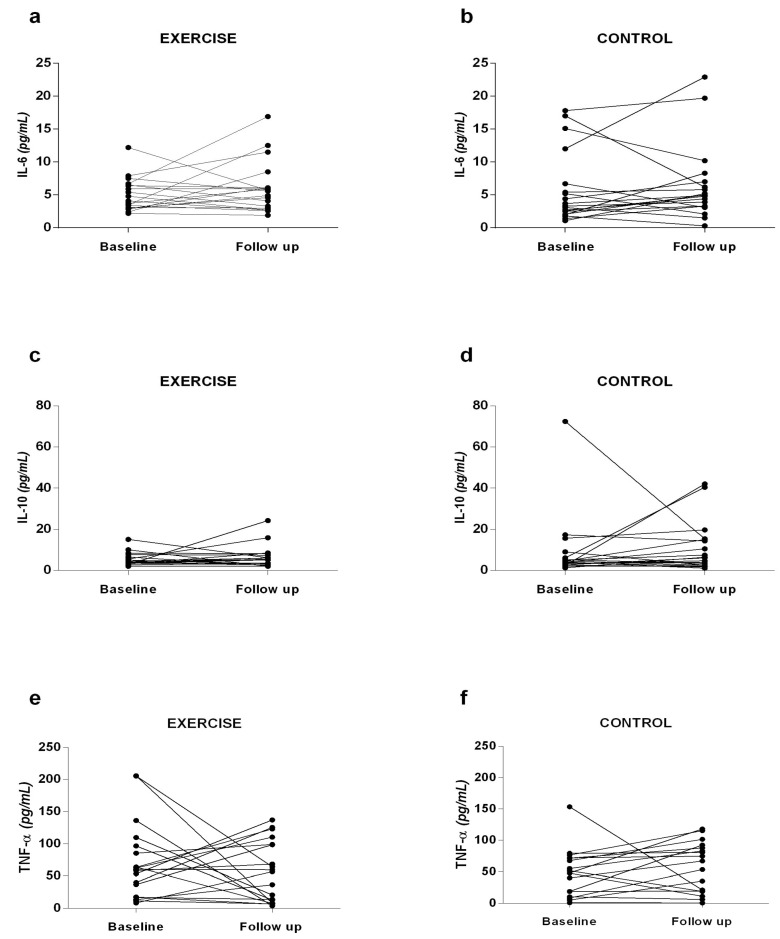
Inflammatory profile. (**a**) IL-6 responses in the Exercise group; (**b**) IL-6 responses in the Control group; (**c**) IL-10 responses in the Exercise group; (**d**) IL-10 responses in the Control group; (**e**) TNF-α responses in the Exercise group; (**f**) TNF-α responses in the Control group.

**Figure 4 ijerph-22-01257-f004:**
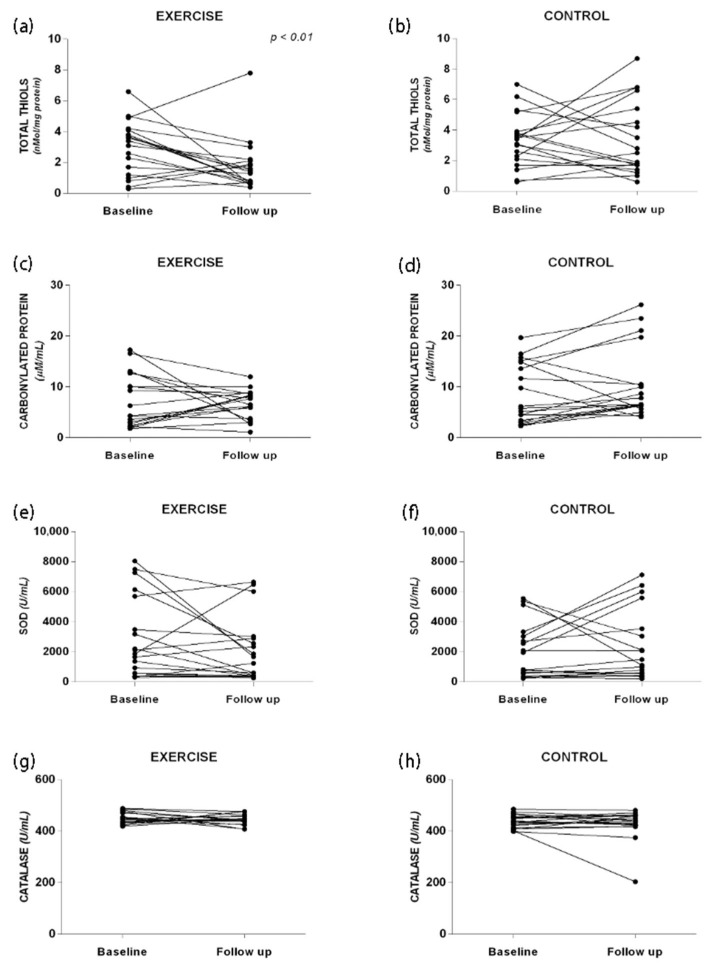
Oxidative stress. (**a**) Total THIOLS responses in the Exercise group; (**b**) Total THIOLS responses in the Control group; (**c**) Carbonylated Protein responses in the Exercise group; (**d**) Carbonylated Protein responses in the Control group; (**e**) SOD responses in the Exercise group; (**f**) SOD responses in the Control group; (**g**) Catalase responses in the Exercise group; (**h**) Catalase responses in the Control group.

**Figure 5 ijerph-22-01257-f005:**
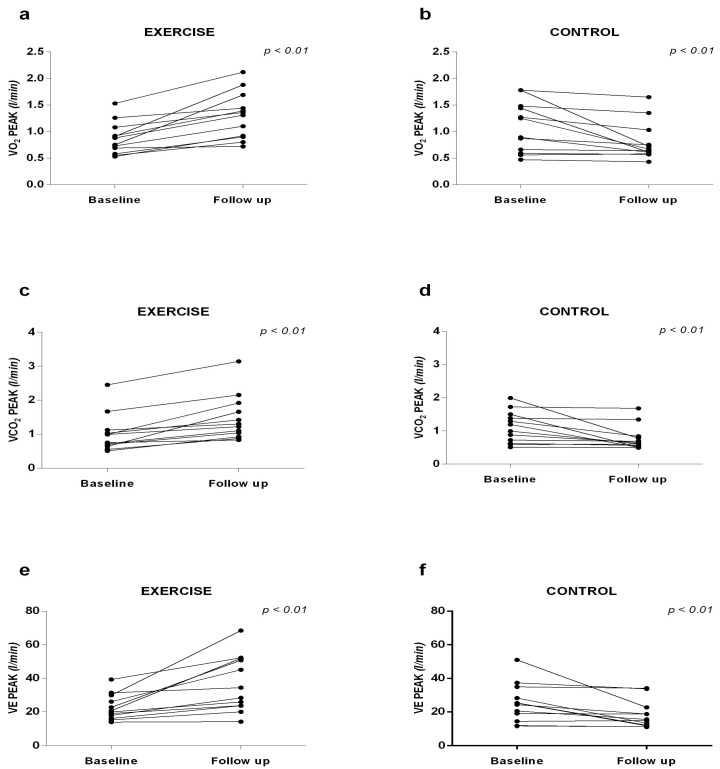
Cardiorespiratory capacity. (**a**) VO_2_Peak responses in the Exercise group; (**b**) VO_2_Peak responses in the Control group; (**c**) VCO_2_Peak responses in the Exercise group; (**d**) VCO_2_Peak responses in the Control group; (**e**) VEPeak responses in the Exercise group; (**f**) VEPeak in the Control group.

**Table 1 ijerph-22-01257-t001:** Baseline sample characteristics.

	Exercise Group	Control Group	*p* Value
Gender, *n* (female, male)	12/7	12/7	1.00
Age, *years*	58.9 ± 10.1	59.6 ± 8.8	0.35
Weight, *Kg*	75.8 ± 17.3	76.4 ± 14.8	0.37
Height, *cm*	158 ± 12	162 ± 8	0.51
BMI, *Kg*/*m^2^*	29.9 ± 5.3	27.9 ± 4.5	0.11
Etiology of Cirrhosis, *n*			
Hepatitis C Virus	10	10	1.00
Non-Alcoholic Steatohepatitis	8	8	1.00
Alcohol	1	1	1.00
Child–Pugh class, *n*			
Child–Pugh A	19	18	1.00
Child-Pugh B	0	1	1.00
Use of Propranolol *(yes*/*no)*	4/15	9/10	0.91
Previous Ascites *(yes*/*no)*	1/18	2/17	0.60
Previous of Edema *(yes*/*no)*	0/19	0/19	1.00
Variceal Bleeding *(yes*/*no)*	4/13	4/15	1.00
Portal Vein Thrombosis *(yes*/*no)*	2/17	1/18	1.00
Esophageal Varices *(yes*/*no)*	7/12	11/9	0.19
Encephalopathy *(yes*/*no)*	0/19	1/18	0.48
Glucose, *mg*/*dL*	111 ± 26	112 ± 46	0.70
Albumin Serum, *g*/*dL*	4.2 ± 0.4	3.9 ± 0.8	0.13
Prothrombin Time, %	81 ± 14	79 ± 20	0.80
INR	1.14 ± 0.13	1.19 ± 0.15	0.37
Serum Bilirubin, *mg*/*dL*	1.03 ± 1.05	1.12 ± 1.09	0.81
AST, *U*/*L*	52 ± 35	41 ± 19	0.18
ALT, *U*/*L*	56 ± 36	45 ± 17	0.60

Legend: BMI, body mass index; INR, International Normalized Ratio; AST, aspartate aminotransferase; ALT, alanine aminotransferase.

**Table 2 ijerph-22-01257-t002:** Blood analysis and body composition.

	Exercise Group			Control Group		
	Baseline	Follow-Up	*p* Value	Baseline	Follow-Up	*p* Value
*Body Composition*						
BMI, *Kg*/*m^2^*	30.0 ± 5.3	29.5 ± 5.0 *	<0.01 *	28.0 ± 4.5	28.1 ± 5.1	0.40
Lean Mass, *kg*	27.7 ± 7.6	27.0 ± 7.0	0.66	27.1 ± 7.1	27.1 ± 6.1	0.90
Fat Mass, *Kg*	28.2 ± 10.6	27.4 ± 10.7	0.33	25.5 ± 9.0	25.4 ± 9.5	0.91

Legend: BMI, body mass index. * Statistical difference from baseline.

**Table 3 ijerph-22-01257-t003:** Muscle strength, total training volume, and exercise capacity.

	Exercise Group			Control Group		
	Baseline	Follow-Up	*p* Value	Baseline	Follow-Up	*p* Value
*Muscle Strength*						
Handgrip, *Kgf*	27.4 ± 13.9	30.0 ± 15.5	<0.01 *	25.6 ± 9.2	21.3 ± 9.0	<0.01 *
*Total Training Volume*						
Lat Pull-down	464 ± 227	862 ± 320	<0.01 *	-	-	
Leg Press	615 ± 270	1297 ± 443	<0.01 *	-	-	
Bench Press	248 ± 153	416 ± 264	<0.01 *	-	-	
Leg Extension	558 ± 359	1080 ± 491	<0.01 *	-	-	
Low Row	473 ± 226	826 ± 326	<0.01 *	-	-	
Leg Curl	530 ± 314	1051 ± 371	<0.01 *	-	-	
Biceps Curl	701 ± 280	1165 ± 336	<0.01 *	-	-	
Tríceps Pulley	445 ± 283	786 ± 546	<0.01 *	-	-	
*Exercise Capacity*						
6 min Walk Test, *m*	456 ± 95	525 ± 95	<0.01	476 ± 51	461 ± 75	0.18

Legend: BMI, body mass index; INR, International Normalized Ratio; AST, aspartate aminotransferase; ALT, alanine aminotransferase. * Statistical difference form baseline.

**Table 4 ijerph-22-01257-t004:** Quality-of-life analysis.

	Exercise Group			Control Group		
	Baseline	Follow-Up	*p* Value	Baseline	Follow-Up	*p* Value
*Quality of Life*						
Physical Function	74 ± 23	90 ± 8	<0.01 *	74 ± 19	71 ± 20	0.49
Role Physical	59 ± 41	96 ± 12	<0.01 *	58 ± 47	61 ± 44	0.72
Bodily Pain	52 ± 23	70 ± 14	<0.01 *	52 ± 27	61 ± 23	0.16
General Health	64 ± 23	76 ± 19	<0.01 *	64 ± 17	65 ± 17	0.87
Vitality	68 ± 23	80 ± 16	0.06	56 ± 23	60 ± 24	0.45
Social Function	81 ± 23	95 ± 10	0.03 *	65 ± 36	68 ± 30	0.67
Role Emotional	54 ± 43	91 ± 27	<0.01 *	47 ± 47	56 ± 47	0.39
Mental Health	71 ± 15	81 ± 13	0.03 *	62 ± 25	64 ± 30	0.45

Legend: * Statistical difference from baseline.

## Data Availability

The data presented in this study are available on request from the corresponding author.

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
