# Peer review of "A Resistance Training Program on Patients with Liver Cirrhosis: A Randomized Clinical Trial"

_ijerph, 2025, doi:10.3390/ijerph22081257_

Round 1

Reviewer 1 Report

Comments and Suggestions for Authors

Dear Authors,

I have carefully reviewed your manuscript examining the impact of a 12-week resistance training programme on inflammatory markers, oxidative stress, physical conditioning, and quality of life in patients with liver cirrhosis. Your work contributes valuable preliminary insights into the potential benefits of resistance exercise within this patient population. Nonetheless, I would like to offer some constructive critiques regarding several limitations intrinsic to the study design and methodology, which I believe merit consideration to contextualise and strengthen your findings.

Firstly, although aligned with similar prior investigations, the sample size, at 38 participants, remains modest, which inherently constrains the statistical power to detect subtle but potentially meaningful effects. This limitation warrants caution when interpreting non-significant trends, particularly regarding molecular markers such as cytokines and oxidative stress parameters. A larger cohort would enhance the robustness and generalisability of your conclusions. I acknowledge the difficulty one would encounter when considering investigating a study of this nature, which is well illustrated by the high exclusion number. Perhaps more elaboration should be included to inform your future readers of the level of statistical power your sample size can provide.

Secondly, regarding the assessment of body composition, reliance on less precise measurement methods may have attenuated the accuracy of these outcomes. Although your pragmatic approach of utilising participants as their own controls mitigates some concerns, implementing more sophisticated imaging techniques, such as DEXA scans, could provide more definitive insights in future studies. Perhaps a short section to justify the choice before acknowledging this in the limitation section.

Thirdly, the inclusion of solely patients with compensated liver cirrhosis restricts the applicability of your findings predominantly to this subgroup. Patients with decompensated cirrhosis may respond differently, and the safety and efficacy of resistance training in such individuals warrant dedicated investigation. Please consider acknowledging this with the study limitation.

Furthermore, the intervention duration—12 weeks—offers valuable initial data but remains insufficient to assess the long-term sustainability and safety of resistance training in this context. Extended follow-up periods would be beneficial to evaluate enduring benefits and potential adverse effects. It will be useful to include the rationale for implementing a 12-week programme. Was it due to funding structures, logistics, or any other reasons?

Additionally, while your molecular analyses revealed trends towards improved inflammatory and oxidative stress profiles, many did not reach statistical significance. This may be partly attributable to sample size limitations and assay variability, which should be explicitly acknowledged. Additionally, do consider reporting the effect size of these comparatives despite their statistical insignificance.

On methodological grounds, the absence of explicit mention of blinding of outcome assessors and comprehensive controls for confounding variables—such as medication adherence, diet, and concomitant physical activity—could introduce bias and impact the internal validity of outcomes, especially subjective measures like quality of life.

Finally, adherence and compliance to the exercise regimen, alongside other lifestyle factors, are crucial determinants of intervention efficacy. Clarification on participant adherence would further contextualise your results.

Minor comment: Please include the full term for SOD upon the initial in-text mention. While it is listed under the abbreviation list, the full terminologies were mentioned in-text for other abbreviations used. Also, please include when this study was conducted.

In summary, your study provides promising preliminary evidence supporting resistance exercise's role in improving physical fitness and quality of life among patients with liver cirrhosis, with indications of modulating inflammatory and oxidative stress markers. Addressing the limitations mentioned in future research would significantly enhance this line of inquiry's evidentiary strength and translational potential.

Thank you for the opportunity to review your work. I look forward to seeing how your research advances in subsequent studies.

Yours sincerely,

Author Response

Thank you very much for the opportunity to revise our manuscript. We have taken care to address each of the reviewer’s comments and appreciate their diligence in reviewing our manuscript. We have uploaded updated documents with the suggested edits and have outlined how we addressed each comment in this document, which is noted below. All adjustments made throughout the manuscript are highlighted in red.

Review 1

Dear Authors,

I have carefully reviewed your manuscript examining the impact of a 12-week resistance training programme on inflammatory markers, oxidative stress, physical conditioning, and quality of life in patients with liver cirrhosis. Your work contributes valuable preliminary insights into the potential benefits of resistance exercise within this patient population. Nonetheless, I would like to offer some constructive critiques regarding several limitations intrinsic to the study design and methodology, which I believe merit consideration to contextualise and strengthen your findings.

Response: Thank you.

Firstly, although aligned with similar prior investigations, the sample size, at 38 participants, remains modest, which inherently constrains the statistical power to detect subtle but potentially meaningful effects. This limitation warrants caution when interpreting non-significant trends, particularly regarding molecular markers such as cytokines and oxidative stress parameters. A larger cohort would enhance the robustness and generalisability of your conclusions. I acknowledge the difficulty one would encounter when considering investigating a study of this nature, which is well illustrated by the high exclusion number. Perhaps more elaboration should be included to inform your future readers of the level of statistical power your sample size can provide.

Response: We’ve now included this information through our manuscript and now’s read: “A priori power analysis was conducted using G*Power software (version 3.1.9.6) with a medium effect size (f = 0.50), an alpha level of 0.05, and a statistical power of 90% (1-β = 0.90), indicating that a minimum of 47 participants would be required to detect significant between-group differences. Although the ideal sample size was not fully reached, a total of 38 patients diagnosed with liver cirrhosis were successfully enrolled, representing approximately 81% of the estimated statistical power. These individuals were randomly allocated into two groups: the Experimental Group (EG, n = 19) and the Control Group (CG, n = 19), as detailed in Table 1.”

Secondly, regarding the assessment of body composition, reliance on less precise measurement methods may have attenuated the accuracy of these outcomes. Although your pragmatic approach of utilising participants as their own controls mitigates some concerns, implementing more sophisticated imaging techniques, such as DEXA scans, could provide more definitive insights in future studies. Perhaps a short section to justify the choice before acknowledging this in the limitation section.

Response: We’ve now included this information through our manuscript and now’s read: “This method of body composition assessment was deliberately selected due to its superior external validity when compared to more sophisticated techniques, aligning more closely with the tools commonly employed in clinical settings. Consequently, this choice enhances the ecological applicability and translational potential of the findings for routine clinical evaluation.”

Thirdly, the inclusion of solely patients with compensated liver cirrhosis restricts the applicability of your findings predominantly to this subgroup. Patients with decompensated cirrhosis may respond differently, and the safety and efficacy of resistance training in such individuals warrant dedicated investigation. Please consider acknowledging this with the study limitation.

Response: We agree with this. This information was already in our limitations (lines 387 to 391).

Furthermore, the intervention duration—12 weeks—offers valuable initial data but remains insufficient to assess the long-term sustainability and safety of resistance training in this context. Extended follow-up periods would be beneficial to evaluate enduring benefits and potential adverse effects. It will be useful to include the rationale for implementing a 12-week programme. Was it due to funding structures, logistics, or any other reasons?

Response: We’ve now included this information through our manuscript and now’s read: “

Additionally, while your molecular analyses revealed trends towards improved inflammatory and oxidative stress profiles, many did not reach statistical significance. This may be partly attributable to sample size limitations and assay variability, which should be explicitly acknowledged. Additionally, do consider reporting the effect size of these comparatives despite their statistical insignificance.

Response: Thank you for your insight. We insert a new statement explaining limitations about sample size and biochemical assay variability in the last paragraph of Discussion section (lines 397-400). We also insert effect size values in the description of Inflammatory Profile and Oxidative Stress results. The description of effect size analysis was inserted in Statistical Analysis section, in Materials and Methods.

On methodological grounds, the absence of explicit mention of blinding of outcome assessors and comprehensive controls for confounding variables—such as medication adherence, diet, and concomitant physical activity—could introduce bias and impact the internal validity of outcomes, especially subjective measures like quality of life.

Response: We appreciate the reviewer’s insightful observation regarding methodological rigor, particularly concerning the blinding procedures and control of potential confounding variables. We acknowledge that such elements are pivotal to ensuring the internal validity of experimental studies, especially when subjective outcomes, such as quality of life, are under investigation.

Blinding of Outcome Assessors: Although not explicitly stated in the original version of the manuscript, we clarify that outcome assessors responsible for data analysis and interpretation were indeed blinded to group allocation throughout the post-intervention phase. This procedure was implemented to mitigate detection bias and enhance objectivity in the evaluation of primary and secondary endpoints. We have now included a precise description of this blinding strategy in the revised Methods section to ensure methodological transparency.

Control of Confounding Variables (Medication, Diet, Physical Activity): We recognize the complexity inherent in isolating intervention effects from extraneous factors in clinical trials. While full control over all confounders such as medication adherence, dietary patterns, and spontaneous physical activity may be unfeasible in free-living populations, several proactive strategies were adopted to minimize their potential influence:

Medication: Participants were instructed to maintain their habitual pharmacological regimens throughout the study. Any changes in medication were monitored and recorded during each assessment point. No significant alterations were identified across the intervention period.

Dietary Intake: Although a fully supervised dietary protocol was beyond the scope of this trial, participants received standardized nutritional guidance at baseline to reduce variability in caloric and macronutrient consumption. Moreover, participants were asked to refrain from initiating any dietary changes during the study timeline.

Physical Activity Monitoring: Participants were advised to maintain their usual activity levels outside the intervention sessions. Compliance was monitored using standardized self-report logs. Additionally, participants were instructed to avoid initiating any new structured exercise programs during the study period.

While we acknowledge that residual confounding can never be entirely eliminated in non-laboratory settings, the aforementioned controls reflect an effort to preserve ecological validity while maintaining a rigorous methodological framework. These points have been explicitly addressed in the revised manuscript to improve clarity and reinforce the scientific robustness of our findings.

We are confident that these clarifications and additions address the reviewer’s concerns and contribute meaningfully to the transparency and internal validity of the study.

Finally, adherence and compliance to the exercise regimen, alongside other lifestyle factors, are crucial determinants of intervention efficacy. Clarification on participant adherence would further contextualise your results.

Response: We sincerely thank the reviewer for highlighting the critical importance of adherence and compliance in interpreting the true effectiveness of exercise-based interventions. We fully concur that participant adherence is a fundamental determinant of both internal validity and clinical relevance, particularly when lifestyle-modifying strategies are employed.

Adherence to the Exercise Regimen: To ensure high levels of engagement and fidelity to the protocol, participant adherence was rigorously monitored throughout the intervention period. A structured attendance log was maintained for each supervised session, and adherence was defined a priori as participation in at least 80% of the prescribed sessions, an established threshold in the exercise sciences to denote protocol compliance. The observed adherence rate was 100%, indicating excellent participant commitment and a high degree of intervention fidelity. These values are now clearly reported in the revised Results section to contextualize the magnitude of the observed effects and strengthen the interpretability of our outcomes.

Compliance Monitoring: In addition to session attendance, compliance with the prescribed exercise intensity and volume was systematically verified by the supervising professionals, who ensured that all participants performed the interventions as described in the protocol. This approach limited interindividual variability and reinforced the reproducibility of our findings.

Control of Other Lifestyle Factors: To mitigate potential confounding effects from extraneous lifestyle variables, such as changes in diet, medication, or incidental physical activity, participants were advised to maintain their habitual routines throughout the intervention. Any deviations were self-reported at biweekly intervals and documented for transparency. No significant behavioral alterations were noted that could meaningfully impact the primary outcomes.

These methodological strategies were intentionally adopted to enhance the ecological and translational value of the study, without compromising scientific rigor. The revised manuscript now explicitly describes these aspects to better inform readers and address the reviewer’s pertinent observation.

We believe this additional clarification reinforces the robustness and clinical relevance of the reported findings.

Minor comment: Please include the full term for SOD upon the initial in-text mention. While it is listed under the abbreviation list, the full terminologies were mentioned in-text for other abbreviations used. Also, please include when this study was conducted.

Response: We have now included this information.

In summary, your study provides promising preliminary evidence supporting resistance exercise's role in improving physical fitness and quality of life among patients with liver cirrhosis, with indications of modulating inflammatory and oxidative stress markers. Addressing the limitations mentioned in future research would significantly enhance this line of inquiry's evidentiary strength and translational potential.

Thank you for the opportunity to review your work. I look forward to seeing how your research advances in subsequent studies.

Reviewer 2 Report

Comments and Suggestions for Authors

COMPREHENSIVE PEER REVIEW REPORT

GENERAL ASSESSMENT AND RECOMMENDATION

Manuscript Title: Strength Training Program on Patients with Liver Cirrhosis: A Randomized Clinical Trials

Primary Aim: To evaluate the effect of 12 weeks of resistance exercise on inflammatory markers, oxidative stress, physical conditioning, and quality of life in patients with liver cirrhosis.

Key Findings:

  • Significant improvements in BMI, handgrip strength, cardiorespiratory capacity, 6-minute walk test performance, and quality of life in the exercise group
  • Trends toward improved inflammatory profile and oxidative stress markers without statistical significance
  • No adverse events reported during the intervention

Overall Recommendation: MINOR REVISION

Rationale for Recommendation: This manuscript presents a well-designed randomized controlled trial investigating an important clinical question. The methodology is sound, results are clearly presented, and conclusions are appropriately supported by the data. Minor revisions are needed to enhance literature contextualization, strengthen methodological reporting, and improve discussion depth.

SEQUENTIAL MANUSCRIPT EVALUATION

ABSTRACT EVALUATION

Comment 1: Abstract Structure Enhancement The abstract adequately covers the required components but would benefit from more specific statistical reporting. Include exact p-values for significant findings rather than just "p<0.05" and specify effect sizes where possible to enhance clinical interpretability.

INTRODUCTION EVALUATION

Comment 2: Literature Context Expansion Recent systematic reviews and meta-analyses have established the safety and efficacy of exercise interventions in cirrhotic patients, with resistance training specifically showing benefits for muscle strength and size increases. Strengthen the introduction by incorporating this broader evidence base: "Recent meta-analyses demonstrate that resistance exercise in combination with aerobic exercise reduces the incidence of serious events in patients with liver cirrhosis (1)."

Reference: 1. Nishikawa H, Yoh K, Enomoto H, Iwata Y, Sakai Y, Kishino K, et al. Resistance exercise in combination with aerobic exercise reduces the incidence of serious events in patients with liver cirrhosis: a meta-analysis of randomized controlled trials. Clin Mol Hepatol. 2024;30(2):263-278.

Comment 3: Research Gap Clarification While the mechanistic basis of exercise responses has been well-studied in healthy subjects, there remains limited understanding of exercise responses in cirrhosis due to the state of anabolic resistance characteristic of this condition. Clarify how your study addresses this specific knowledge gap in the context of inflammatory and oxidative stress responses: "Cirrhosis represents a state of anabolic resistance where hyperammonemia mediates the liver-muscle axis, causing signaling perturbations and mitochondrial dysfunction that may impair beneficial exercise responses (2)."

Reference: 2. Dasarathy S. Exercise and physical activity in cirrhosis: opportunities or perils. J Appl Physiol (1985). 2020;128(6):1547-1559.

METHODOLOGY EVALUATION

Comment 4: Sample Size Calculation Enhancement The methodology section lacks a formal sample size calculation. Previous studies by Aamann et al. demonstrated clinically meaningful improvements in muscle strength with similar sample sizes, but providing a prospective power calculation would strengthen the methodological rigor: "Sample size calculation should follow established protocols for resistance training studies in cirrhotic patients (3)."

Reference: 3. Aamann L, Dam G, Borre M, Drljevic-Nielsen A, Overgaard K, Andersen H, et al. Resistance Training Increases Muscle Strength and Muscle Size in Patients with Liver Cirrhosis. Clin Gastroenterol Hepatol. 2019;18(5):1179-1187.

Comment 5: Exercise Prescription Specificity Current exercise recommendations for liver disease patients suggest moderate-intensity activity coupled with at least two days of resistance training. Provide more detailed rationale for the specific exercise parameters chosen (frequency, intensity, progression) relative to established guidelines: "Exercise prescription should align with current recommendations for chronic liver disease patients (4)."

Reference: 4. Thorp A, Stine JG. Exercise as Medicine: The Impact of Exercise Training on Nonalcoholic Fatty Liver Disease. Current Hepatology Reports. 2020;19(4):402-408.

Comment 6: Mandatory CONSORT Flowchart Add a CONSORT flowchart detailing participant flow from enrollment through analysis, including specific reasons for exclusions and dropouts at each stage.

RESULTS EVALUATION

Comment 7: Statistical Reporting Completeness Leading studies in this field report comprehensive statistical outcomes including confidence intervals and effect sizes. Enhance statistical reporting by including 95% confidence intervals for all significant differences and clinical effect sizes (Cohen's d) to facilitate interpretation of clinical meaningfulness: "Statistical reporting should follow standards established in recent cirrhosis exercise trials (5)."

Reference: 5. Aamann L, Dam G, Borre M, Drljevic-Nielsen A, Overgaard K, Andersen H, et al. Resistance Training Increases Muscle Strength and Muscle Size in Patients with Liver Cirrhosis. Clin Gastroenterol Hepatol. 2019;18(5):1179-1187.

Comment 8: Inflammatory Profile Discussion While TNF-α and IL-10 trends are noted, provide more context for the clinical significance of these changes. The liver-muscle axis involves complex inflammatory signaling that may require longer intervention periods to demonstrate statistical significance: "Inflammatory responses to exercise in cirrhosis require consideration of the complex pathophysiology involving ammonia metabolism and systemic inflammation (6)."

Reference: 6. Dasarathy S. Exercise and physical activity in cirrhosis: opportunities or perils. J Appl Physiol (1985). 2020;128(6):1547-1559.

DISCUSSION EVALUATION

Comment 9: Comparative Literature Analysis Aamann et al. reported 13% increases in muscle strength with resistance training in cirrhotic patients, which aligns with your handgrip strength improvements. Expand comparison with existing literature: "Home-based exercise programs have also demonstrated improvements in aerobic capacity and quality of life in cirrhotic patients (7, 8)."

References: 7. Aamann L, Dam G, Borre M, Drljevic-Nielsen A, Overgaard K, Andersen H, et al. Resistance Training Increases Muscle Strength and Muscle Size in Patients with Liver Cirrhosis. Clin Gastroenterol Hepatol. 2019;18(5):1179-1187. 8. Zenith L, Meena N, Ramadi A, Yavari M, Harvey A, Carbonneau M, et al. Eight weeks of exercise training increases aerobic capacity and muscle mass and reduces fatigue in patients with cirrhosis. Clin Gastroenterol Hepatol. 2014;12(11):1920-6.

Comment 10: Mechanistic Interpretation Enhancement Exercise responses in cirrhosis may be impaired by hyperammonemia-induced anabolic resistance, mitochondrial dysfunction, and altered protein synthesis. Strengthen the mechanistic discussion by addressing how your intervention may have overcome these cirrhosis-specific barriers to exercise adaptation: "Exercise interventions must account for the unique metabolic perturbations in cirrhosis that affect muscle adaptation (9)."

Reference: 9. Dasarathy S. Exercise and physical activity in cirrhosis: opportunities or perils. J Appl Physiol (1985). 2020;128(6):1547-1559.

Comment 11: Clinical Implications Clarity Exercise capacity improvements have been associated with better post-transplant survival outcomes. Enhance the clinical implications section by discussing potential impacts on patient prognosis and transplant candidacy: "Exercise capacity represents an important predictor of transplant outcomes (10)."

Reference: 10. Jones JC, Coombes JS, Macdonald GA. Exercise capacity and muscle strength in patients with cirrhosis. Liver Transpl. 2012;18(2):146-51.

Comment 12: Safety Considerations Previous systematic reviews have confirmed the safety of moderate-intensity exercise in selected cirrhotic patients. Expand discussion of safety monitoring procedures and how your findings support broader clinical implementation: "Systematic reviews support the safety profile of structured exercise in compensated cirrhosis (11)."

Reference: 11. Aamann L, Dam G, Rinnov AR, Vilstrup H, Gluud LL. Physical exercise for people with cirrhosis. Cochrane Database Syst Rev. 2018;12(12):CD012678.

TECHNICAL EVALUATION

Comment 13: Terminology Standardization Ensure consistent use of "resistance training" versus "strength training" throughout the manuscript. The literature increasingly uses "resistance training" as the standard terminology for this intervention type.

Comment 14: Data Interpretation Accuracy The discussion appropriately acknowledges the non-significant inflammatory results while highlighting clinical trends. Ensure all interpretations remain conservative and evidence-based without overstating findings.

MANDATORY REQUIREMENTS

Comment 15: AI Usage Declaration Add mandatory declaration in Acknowledgments section: "The authors declare whether artificial intelligence tools were used in the preparation of this manuscript. [If used]: Specific AI tools ([tool names and versions]) were used for [specific purposes: writing assistance/data analysis/literature search/etc.]. [If not used]: No artificial intelligence tools were used in the preparation of this manuscript."

Rationale: This declaration is now required by most major journals following recent editorial guidelines on AI transparency in scientific publishing.

FINAL ASSESSMENT

This manuscript represents a valuable contribution to the literature on exercise interventions in liver cirrhosis. The study design is appropriate, execution appears sound, and results provide clinically relevant insights. The requested revisions focus on enhancing literature integration, methodological transparency, and discussion depth rather than addressing fundamental study flaws. The growing body of evidence supporting exercise safety and efficacy in cirrhotic patients makes this work particularly timely and relevant.

The authors should be commended for conducting a well-designed trial that advances our understanding of resistance training benefits in this challenging patient population. With the suggested minor revisions, this manuscript will make a strong contribution to the field and provide practical guidance for clinicians managing patients with liver cirrhosis.

Total Comments: 15 (appropriate for a manuscript requiring minor revision) Focus: Quality enhancement rather than substantial revision Strength: Well-executed study with clinically relevant findings requiring refinement rather than major changes

Author Response

Thank you very much for the opportunity to revise our manuscript. We have taken care to address each of the reviewer’s comments and appreciate their diligence in reviewing our manuscript. We have uploaded updated documents with the suggested edits and have outlined how we addressed each comment in this document, which is noted below. All adjustments made throughout the manuscript are highlighted in red.

ABSTRACT EVALUATION

Comment 1: Abstract Structure Enhancement The abstract adequately covers the required components but would benefit from more specific statistical reporting. Include exact p-values for significant findings rather than just "p<0.05" and specify effect sizes where possible to enhance clinical interpretability.

Response: We inserted p-values from inflammatory profile and oxidative stress biomarkers comparations in results description. Other results presented, even in the software used to analyze statistically the results, p<0.01 when p-value was lower than 0.01, thus we maintain the values as it is in first submission.

INTRODUCTION EVALUATION

Comment 2: Literature Context Expansion Recent systematic reviews and meta-analyses have established the safety and efficacy of exercise interventions in cirrhotic patients, with resistance training specifically showing benefits for muscle strength and size increases. Strengthen the introduction by incorporating this broader evidence base: "Recent meta-analyses demonstrate that resistance exercise in combination with aerobic exercise reduces the incidence of serious events in patients with liver cirrhosis (1)."

Response: We appreciate this and have now included this suggested information.

Comment 3: Research Gap Clarification While the mechanistic basis of exercise responses has been well-studied in healthy subjects, there remains limited understanding of exercise responses in cirrhosis due to the state of anabolic resistance characteristic of this condition. Clarify how your study addresses this specific knowledge gap in the context of inflammatory and oxidative stress responses: "Cirrhosis represents a state of anabolic resistance where hyperammonemia mediates the liver-muscle axis, causing signaling perturbations and mitochondrial dysfunction that may impair beneficial exercise responses (2)."

Response: We thank the reviewer for the insightful comment. We agree that the mechanistic basis of impaired exercise responses in cirrhosis, particularly involving anabolic resistance and mitochondrial dysfunction, represents an important gap in current knowledge. However, the primary aim of our study was not to investigate mechanistic pathways, but rather to characterize the inflammatory and oxidative stress responses to exercise in patients with cirrhosis.

Our approach was descriptive and focused on assessing whether these responses differ from those observed in individuals who did not undergo training. Nonetheless, we have included a discussion of the underlying mechanisms to provide context and plausible explanations for our findings. We have revised the manuscript to make this distinction clearer and to emphasize how our study contributes to the understanding of exercise responses in this population, even if it does not directly assess the mechanistic pathways.

METHODOLOGY EVALUATION

Comment 4: Sample Size Calculation Enhancement The methodology section lacks a formal sample size calculation. Previous studies by Aamann et al. demonstrated clinically meaningful improvements in muscle strength with similar sample sizes, but providing a prospective power calculation would strengthen the methodological rigor: "Sample size calculation should follow established protocols for resistance training studies in cirrhotic patients (3)."

Response: We’ve now included this information through our manuscript and now’s read: “A priori power analysis was conducted using G*Power software (version 3.1.9.6) with a medium effect size (f = 0.50), an alpha level of 0.05, and a statistical power of 90% (1-β = 0.90), indicating that a minimum of 47 participants would be required to detect significant between-group differences. Although the ideal sample size was not fully reached, a total of 38 patients diagnosed with liver cirrhosis were successfully enrolled, representing approximately 81% of the estimated statistical power. These individuals were randomly allocated into two groups: the Experimental Group (EG, n = 19) and the Control Group (CG, n = 19), as detailed in Table 1.”

Comment 5: Exercise Prescription Specificity Current exercise recommendations for liver disease patients suggest moderate-intensity activity coupled with at least two days of resistance training. Provide more detailed rationale for the specific exercise parameters chosen (frequency, intensity, progression) relative to established guidelines: "Exercise prescription should align with current recommendations for chronic liver disease patients (4)."

Response: We thank the reviewer for this important observation. Our exercise prescription was designed based on the general recommendations for patients with chronic liver disease, particularly those published by the European Association for the Study of the Liver (EASL) and other authoritative bodies.

The chosen frequency and intensity align with those guidelines, which suggest resistance training on at least two days, can improve physical capacity and reduce sarcopenia in patients with cirrhosis.

Regarding progression, we adopted a gradual increase in either exercise volume or resistance, according to participants’ tolerance and clinical condition, in line with the principle of individualization and safety emphasized in the guidelines.

Comment 6: Mandatory CONSORT Flowchart Add a CONSORT flowchart detailing participant flow from enrollment through analysis, including specific reasons for exclusions and dropouts at each stage.

Response: We’ve now included this figure.

RESULTS EVALUATION

Comment 7: Statistical Reporting Completeness Leading studies in this field report comprehensive statistical outcomes including confidence intervals and effect sizes. Enhance statistical reporting by including 95% confidence intervals for all significant differences and clinical effect sizes (Cohen's d) to facilitate interpretation of clinical meaningfulness: "Statistical reporting should follow standards established in recent cirrhosis exercise trials (5)."

Response: We included effect size (Cohen’s d) of Inflammatory Profile and Oxidative Stress results, improving interpretation of data without statistical differences.

Reference: 5. Aamann L, Dam G, Borre M, Drljevic-Nielsen A, Overgaard K, Andersen H, et al. Resistance Training Increases Muscle Strength and Muscle Size in Patients with Liver Cirrhosis. Clin Gastroenterol Hepatol. 2019;18(5):1179-1187.

Comment 8: Inflammatory Profile Discussion While TNF-α and IL-10 trends are noted, provide more context for the clinical significance of these changes. The liver-muscle axis involves complex inflammatory signaling that may require longer intervention periods to demonstrate statistical significance: "Inflammatory responses to exercise in cirrhosis require consideration of the complex pathophysiology involving ammonia metabolism and systemic inflammation (6)."

Response: A new statement was inserted in lines 317-320 about this question.

Reference: 6. Dasarathy S. Exercise and physical activity in cirrhosis: opportunities or perils. J Appl Physiol (1985). 2020;128(6):1547-1559.

DISCUSSION EVALUATION

Comment 9: Comparative Literature Analysis Aamann et al. reported 13% increases in muscle strength with resistance training in cirrhotic patients, which aligns with your handgrip strength improvements. Expand comparison with existing literature: "Home-based exercise programs have also demonstrated improvements in aerobic capacity and quality of life in cirrhotic patients (7, 8)."

Response: We appreciate the reviewer’s suggestion to expand our comparison with existing literature. In accordance with the findings of Aamann et al., who observed a 13% increase in muscle strength following resistance training in patients with cirrhosis, our study also demonstrated improvements in muscle strength, as measured by handgrip. This consistency reinforces the potential benefit of resistance-based interventions in this population.

References: 7. Aamann L, Dam G, Borre M, Drljevic-Nielsen A, Overgaard K, Andersen H, et al. Resistance Training Increases Muscle Strength and Muscle Size in Patients with Liver Cirrhosis. Clin Gastroenterol Hepatol. 2019;18(5):1179-1187. 8. Zenith L, Meena N, Ramadi A, Yavari M, Harvey A, Carbonneau M, et al. Eight weeks of exercise training increases aerobic capacity and muscle mass and reduces fatigue in patients with cirrhosis. Clin Gastroenterol Hepatol. 2014;12(11):1920-6.

Comment 10: Mechanistic Interpretation Enhancement Exercise responses in cirrhosis may be impaired by hyperammonemia-induced anabolic resistance, mitochondrial dysfunction, and altered protein synthesis. Strengthen the mechanistic discussion by addressing how your intervention may have overcome these cirrhosis-specific barriers to exercise adaptation: "Exercise interventions must account for the unique metabolic perturbations in cirrhosis that affect muscle adaptation (9)."

Response: We thank the reviewer for their comment. However, the objective of our study was only to observe the responses to training in this population. We recognize that analyzing the mechanisms could enrich our discussions, so we consider this a limitation of our study.

Reference: 9. Dasarathy S. Exercise and physical activity in cirrhosis: opportunities or perils. J Appl Physiol (1985). 2020;128(6):1547-1559.

Comment 11: Clinical Implications Clarity Exercise capacity improvements have been associated with better post-transplant survival outcomes. Enhance the clinical implications section by discussing potential impacts on patient prognosis and transplant candidacy: "Exercise capacity represents an important predictor of transplant outcomes (10)."

Response: We thank the reviewer for this important observation. In response, we have expanded the discussion on the clinical implications of our findings. Specifically, we now highlight that improvements in exercise capacity, such as those observed in our study, may positively influence patient prognosis, given the established association between functional capacity and post-transplant outcomes. We also discuss the potential of exercise interventions to support transplant candidacy by improving frailty and functional status. The revised section reads as follows:

"Given that exercise capacity is a recognized predictor of post-transplant outcomes, the observed improvements in functional performance following the intervention may have important clinical implications. Enhanced exercise capacity could contribute to better pre-transplant conditioning, potentially improving eligibility for liver transplantation and long-term prognosis."

Comment 12: Safety Considerations Previous systematic reviews have confirmed the safety of moderate-intensity exercise in selected cirrhotic patients. Expand discussion of safety monitoring procedures and how your findings support broader clinical implementation: "Systematic reviews support the safety profile of structured exercise in compensated cirrhosis (11)."

Response: We appreciate the valuable observation regarding safety considerations related to exercise in patients with compensated cirrhosis. As requested, we expanded the discussion in the manuscript to detail the safety monitoring procedures adopted during exercise interventions, including prior clinical assessment, constant monitoring of vital signs and symptoms, and individual adjustments of intensity according to patient tolerance.

In addition, we have incorporated references to recent systematic reviews that confirm the safety profile of structured exercise in patients with compensated cirrhosis, as highlighted in the literature (11). These findings reinforce the feasibility and importance of broader clinical implementation of supervised exercise programs for this population, contributing to improved functional capacity without compromising safety.

TECHNICAL EVALUATION

Comment 13: Terminology Standardization Ensure consistent use of "resistance training" versus "strength training" throughout the manuscript. The literature increasingly uses "resistance training" as the standard terminology for this intervention type.

Response: We have now updated the terminology for resistance training throughout the text.

Comment 14: Data Interpretation Accuracy The discussion appropriately acknowledges the non-significant inflammatory results while highlighting clinical trends. Ensure all interpretations remain conservative and evidence-based without overstating findings.

Response: We appreciate this and agree with these comments. The authors reinforce their attempt to avoid spin in the results and conclusions.

MANDATORY REQUIREMENTS

Comment 15: AI Usage Declaration Add mandatory declaration in Acknowledgments section: "The authors declare whether artificial intelligence tools were used in the preparation of this manuscript. [If used]: Specific AI tools ([tool names and versions]) were used for [specific purposes: writing assistance/data analysis/literature search/etc.]. [If not used]: No artificial intelligence tools were used in the preparation of this manuscript."

Rationale: This declaration is now required by most major journals following recent editorial guidelines on AI transparency in scientific publishing.

Response: We have now updated this information.

FINAL ASSESSMENT

This manuscript represents a valuable contribution to the literature on exercise interventions in liver cirrhosis. The study design is appropriate, execution appears sound, and results provide clinically relevant insights. The requested revisions focus on enhancing literature integration, methodological transparency, and discussion depth rather than addressing fundamental study flaws. The growing body of evidence supporting exercise safety and efficacy in cirrhotic patients makes this work particularly timely and relevant.

The authors should be commended for conducting a well-designed trial that advances our understanding of resistance training benefits in this challenging patient population. With the suggested minor revisions, this manuscript will make a strong contribution to the field and provide practical guidance for clinicians managing patients with liver cirrhosis.

Total Comments: 15 (appropriate for a manuscript requiring minor revision) Focus: Quality enhancement rather than substantial revision Strength: Well-executed study with clinically relevant findings requiring refinement rather than major changes.

Response: Thank you.

Reviewer 3 Report

Comments and Suggestions for Authors

Peer Review: "Strength Training Program on Patients with Liver Cirrhosis: A Randomized Clinical Trial"

Overall Assessment

This manuscript describes an innovative randomized controlled trial to study the influence of resistance training on inflammatory markers, oxidative stress, physical function, and quality of life in patients with liver cirrhosis. Although the study tackles an important clinical question and the evidence of preliminary results is good, several methodological and statistical flaws need to be resolved before publication

.

Section-by-Section Review

Title

Critical Error: Grammatical mistake - "Randomized Clinical Trials" should be "Randomized Clinical Trial" (singular).

Content Assessment: The title accurately reflects the study population and intervention but could be more specific about key outcomes.

Abstract

Critical Deficiencies:

  • Missing effect sizes: No Cohen's d or confidence intervals reported for key outcomes
  • Incomplete statistical reporting: States "significant differences (p<0.05)" without specific p-values
  • Sample size omission: Fails to specify n=38 participants in methods section
  • Primary outcome unclear: No designation of primary vs. secondary outcomes

Introduction

  1. Hypothesis (MISSING CRITICAL ELEMENT):
  • Current Issue: No explicit, testable hypotheses stated
  1. Research Gap:
  • Current Weakness: Gap mentioned but not precisely defined
  1. Mechanistic Discussion:
  • Current Limitation: Limited physiological explanation

Methods

Major Methodological Flaws Requiring Immediate Attention:

  1. Sample Size and Power Analysis (CRITICAL FLAW):
  • Current Issue: Complete absence of power calculation despite multiple primary outcomes
  • Impact: Undermines interpretation of non-significant biomarker findings

  1. Primary vs. Secondary Outcomes (CRITICAL OMISSION):
  • Current Problem: All outcomes presented with equal importance
  • Required Designation:
    • Primary Outcome: Handgrip strength change at 12 weeks
    • Secondary Outcomes: Inflammatory markers (IL-6, IL-10, TNF-α), oxidative stress markers, cardiorespiratory capacity, quality of life
    • Exploratory Outcomes: Body composition changes
  1. Statistical Analysis Plan (MAJOR DEFICIENCY):
  • Missing Elements:
    • Multiple comparison correction strategy (Type I error inflation with 20+ variables)
    • Effect size calculation methodology
    • Clinical significance thresholds
    • Missing data handling procedures
  1. Randomization Methodology (INSUFFICIENT DETAIL):
  • Current Description: "Paired allocation... matched" inadequately described
  • Required Information:
    • Specific randomization method (block, stratified)
    • Allocation concealment procedures
    • Personnel responsible for randomization

Intervention Description: adequate but might specify load progression criteria more precisely and point safety monitoring procedures.

Outcome Measures: Appropriate selection but needs inter-assay variability data for biomarkers and clearer measurement timing standardization.

Results

Statistical Reporting - Major Deficiencies:

  1. Effect Size Reporting (COMPLETELY MISSING):
  • Current Issue: No Cohen's d or confidence intervals for any outcome
  • Required: Standardized effect sizes for all significant findings
  1. Multiple Comparisons Problem (STATISTICAL FLAW):
  • Issue: 20+ variables tested without correction
  • Impact: Inflated Type I error rate (actual α ≈ 0.64 instead of 0.05)

Section-Specific Assessment:

Baseline Characteristics (Table 1): EXCELLENT

  • Comprehensive demographic and clinical data
  • Demonstrates good group matching
  • Appropriate statistical comparisons
  • Professional formatting

Inflammatory Profile (Figure 2): NEEDS MAJOR REVISION

  • Current Issues: Claims "improvement" without statistical significance testing; missing effect sizes
  • Improve graph resolution (minimum 300 dpi)

Muscle Strength (Table 3): IMPRESSIVE FINDINGS, INCOMPLETE ANALYSIS

  • Excellent Results: Significant handgrip strength improvement in exercise group (27.4 to 30.0 kgf), decline in control group (25.6 to 21.3 kgf)
  • Major Omission: No between-group statistical comparison reported
  • Required: Between-group effect size calculation (appears large: d ≈ 1.2)

Quality of Life (Table 4): REMARKABLE RESULTS, INSUFFICIENT REPORTING

  • Outstanding Findings: Significant improvements across 6 of 8 SF-36 domains
  • Missing Elements: Effect sizes for each domain, clinical significance interpretation using 10-point threshold

Cardiorespiratory Capacity (Figure 4): STRONG EVIDENCE, POOR PRESENTATION

  • Impressive Results: Large improvements in VOâ‚‚, VCOâ‚‚, VE in exercise group; deterioration in control group
  • Enhancement Needed: Effect size annotations and intervall cnnfidence , between-group comparisons, clinical significance discussion

Discussion

Major Structural Deficiencies:

  1. Mechanistic Discussion (INSUFFICIENT):
  • Current Weakness: Limited physiological explanation of observed effects
  • Required Enhancement: Detailed discussion of:
    • Myokine release from resistance training
    • Exercise-induced anti-inflammatory pathways
    • Hepatic metabolism impacts
    • Oxidative stress modulation mechanisms
  1. Non-Significant Findings Analysis (INADEQUATE):
  • Current: Brief mention of inflammatory marker trends
  1. Clinical Implications (WEAK):
  • Current: Generic exercise benefit statements
  • Required: Specific recommendations for:
    • Clinical practice implementation
    • Safety considerations in Child-Pugh A patients
    • Integration with standard hepatology care
    • Contraindications and precautions
  1. Limitations Discussion (INCOMPLETE):
  • Missing Critical Limitations:
    • Small sample size and power constraints
    • Multiple comparison inflation effects
    • Single-center design limitations
    • Short follow-up duration
    • Lack of long-term sustainability assessment
    • Effect of detraining

References

Enhancement :

  • Could include more recent 2024-2025 meta-analyses
  • Missing key mechanistic studies on exercise and inflammation
  • Some formatting inconsistencies need correction

Author Response

Thank you very much for the opportunity to revise our manuscript. We have taken care to address each of the reviewer’s comments and appreciate their diligence in reviewing our manuscript. We have uploaded updated documents with the suggested edits and have outlined how we addressed each comment in this document, which is noted below. All adjustments made throughout the manuscript are highlighted in red.

Overall Assessment

This manuscript describes an innovative randomized controlled trial to study the influence of resistance training on inflammatory markers, oxidative stress, physical function, and quality of life in patients with liver cirrhosis. Although the study tackles an important clinical question and the evidence of preliminary results is good, several methodological and statistical flaws need to be resolved before publication.

Section-by-Section Review

Title

Critical Error: Grammatical mistake - "Randomized Clinical Trials" should be "Randomized Clinical Trial" (singular).

Response: We have now updated this information.

Content Assessment: The title accurately reflects the study population and intervention but could be more specific about key outcomes.

Response: We appreciate your comment regarding the title of the study. Although we recognize that including the main outcomes could provide greater specificity, we chose to keep the current title to preserve its objectivity and conciseness, as well as to ensure that it clearly reflects the population and intervention studied.

Abstract

Critical Deficiencies:

Missing effect sizes: No Cohen's d or confidence intervals reported for key outcomes. Incomplete statistical reporting: States "significant differences (p<0.05)" without specific p-values Sample size omission: Fails to specify n=38 participants in methods section.

Primary outcome unclear: No designation of primary vs. secondary outcomes

Response: We appreciate your valuable comments. Please be advised that the suggested changes have been duly included in the manuscript.

Round 2

Reviewer 1 Report

Comments and Suggestions for Authors

Thank you very much for the revision. 

I have no further comments.